# Effect of Thawing Procedure and Thermo-Resistance Test on Sperm Motility and Kinematics Patterns in Two Bovine Breeds

**DOI:** 10.3390/ani14192768

**Published:** 2024-09-25

**Authors:** Juan M. Solís, Francisco Sevilla, Miguel A. Silvestre, Ignacio Araya-Zúñiga, Eduardo R. S. Roldan, Alejandro Saborío-Montero, Anthony Valverde

**Affiliations:** 1Animal Reproduction Laboratory, School of Agronomy, Costa Rica Institute of Technology, San Carlos Campus, Alajuela 223-21002, Costa Rica; jusolis@estudiantec.cr (J.M.S.); f.sevilla@tec.ac.cr (F.S.); igaraya@estudiantec.cr (I.A.-Z.); 2Department of Cellular Biology, Functional Biology and Physical Anthropology, University of Valencia, Campus Burjassot, C/Dr Moliner, 50, 46100 Valencia, Spain; miguel.silvestre@uv.es; 3Department of Biodiversity and Evolutionary Biology, National Museum of Natural Sciences, Spanish National Research Council (CSIC), 28006 Madrid, Spain; roldane@mncn.csic.es; 4Alfredo Volio Mata Experimental Station, Faculty of Agri-Food Sciences, University of Costa Rica, Cartago 30304, Costa Rica; alejandro.saboriomontero@ucr.ac.cr

**Keywords:** sperm kinematics, reproductive biotechnology, cryopreservation, artificial insemination

## Abstract

**Simple Summary:**

Artificial insemination (AI) is crucial for dairy cattle worldwide due to its impact on reproductive efficiency and profitability. This study examined the effects of thawing time and temperature, combined with a thermo-resistance test (TRT), on sperm motility and kinematic variables in dairy bulls’ semen for AI. The semen thawing process is critical in maintaining sperm quality, requiring strict adherence to standardized techniques. Incorrect thawing procedures can damage sperm structures. Rapid thawing is necessary to prevent ice recrystallization, and excessive heat exposure can lead to detrimental changes. The thermo-resistance test simulates conditions of the female reproductive tract to assess sperm longevity. Results indicated that Jersey bulls were greater for total and progressive motility percentages compared to Holstein bulls. Jersey bulls exhibited higher values for various velocity parameters and linearity, while Holstein bulls had a lower crossover frequency. The optimal thawing condition was 37 °C for 30 s. Sperm quality decreased with longer post-thawing times.

**Abstract:**

This investigation aimed to analyze the effect that thawing time and temperature in combination with a termo-resistance test had on straws from dairy bulls used for artificial insemination (AI) on semen motility and kinematic variables measured with CASA systems. Eight animals of Holstein and Jersey breeds were used, and nine frozen-thawed semen doses per animal were analyzed for each breed. Three temperatures (35, 37, and 40 °C) and three thawing times (35, 40, and 45 s) were evaluated using a factorial design. Motility and kinematic patterns were analyzed using CASA-mot (Computer-Assisted Semen Analysis of motility) technology at different post-thawing times (0.5, 1, and 2 h). Sperm motility in Jersey bulls was higher (*p* < 0.05) than in Holstein ones (64.52 ± 1.45% and 53.10 ± 1.40%, respectively). The same effect was seen with progressive motility among the two breeds (Jersey: 45.29 ± 1.00%; Holstein: 36.30 ± 0.98%, *p* < 0.05). The Jersey breed presented higher values (*p* < 0.05) of curvilinear velocity (VCL), rectilinear velocity (VSL), average velocity (VAP), linearity on forward progression (LIN), and wobble (WOB). The Holstein breed showed a lower mean value (*p* < 0.05) of the beat-cross frequency (BCF) compared to the Jersey breed, thus suggesting an effect on VCL and VAP. During the post-thaw period, a gradual increase in VCL was observed at 2 h. VSL and VAP showed a decrease (*p* < 0.05) as the post-thaw period was prolonged. The study showed differences in sperm quality between Holstein and Jersey breeds, influenced by cryopreservation, thawing, and post-thawing incubation. Thawing at 37 °C for 30 s was considered optimal in relation to sperm motility. In addition, a decrease in sperm quality was observed as post-thawing time increased.

## 1. Introduction

Reproductive technologies such as artificial insemination (AI) are important practices worldwide for dairy cattle, whose profitability depends directly on their reproductive efficiency [1,2]. This reproductive biotechnology mainly uses frozen-thawed semen doses [3,4,5], which enhances the genetic improvement of dairy cattle, thus implying a real improvement in bovine reproduction [6].

Variations in the sperm quality of cryopreserved materials could be related to genetic factors [7], as it has been observed that some bulls have a superior freezing capacity than others [8]. This genetic variation may directly affect the resistance of spermatozoa to damage due to cryopreservation [9]. In addition, molecular factors, such as the presence of specific proteins associated with cryotolerance, could be an important determinant in the ability of spermatozoa to maintain their viability during the freezing and thawing process [10]. Finally, environmental factors, such as temperature conditions and semen handling during cryopreservation, may also influence the process’s efficiency and quality of semen after thawing. These factors could contribute to the observed variability in sperm response to cryopreservation [11].

The semen thawing process is a critical phase in the management of reproductive material, in which strict compliance with standardized techniques is fundamental to preserve the maximum semen quality [12]. Variations during this process can lead to metabolic decreases in cells, changes in cell membranes, organelles, or gametic interaction, which negatively impacts fertilization ability [2]. Previous research has shown that a proper thawing process improves sperm motility and preserves cellular integrity, allowing progressively motile sperm to generate ATP, maintain plasma membrane and acrosome integrity, and retain enzymes essential for oocyte fertilization [13].

Studies showed the critical importance of an accurate management of temperature and exposure time of frozen semen to maintain sperm structure and function [14]. However, time and temperature protocols for thawing bull semen straws may vary between studies [6,12,15,16]. Thawing procedures can cause significant damage to movement and cellular structures; this damage depends on the rate of cooling during freezing, which requires rapid thawing to prevent problems such as ice recrystallization [17]. In addition, prolonged exposure to high temperatures during this process can lead to detrimental changes in pH and protein denaturation, which can result in cell death [10]. Therefore, it is necessary to establish protocols that allow for reducing the impact on the quality of semen once it has been thawed [17,18,19].

In vitro tests, such as the thermal-resistance test (TRT), may provide additional information to evaluate the effects of the thawing process [20,21]. TRT is an attempt to simulate the time that sperm can spend in the female reproductive tract, allowing the longevity of sperm to be assessed by subjecting them to in vitro incubation at physiological temperature for a given period [22,23]. Indeed, the significant differences in sperm motility observed between thawing protocols disappeared after a TRT [24]. Therefore, this study aimed to evaluate the effect of thawing time and temperature in combination with a TRT on sperm motility and kinematics variables through a CASA-mot system in Holstein and Jersey bulls.

## 2. Materials and Methods

### 2.1. Ethical Approval

This study was approved by the Institutional Review Board following ethical principles and by the Committee of Centro de Investigación y Desarrollo en Agricultura Sostenible para el Trópico Húmedo at the Costa Rica Institute of Technology (CIDASTH-ITCR), according to Section 08/2023, article 5.0, DAGSC-075-2023, and CIE-206-2023 and conducted in accordance with the Three Rs principle.

### 2.2. Study Period and Location

The present study was completed from April to November 2023 at the Animal Reproduction Laboratory (AndroTEC), located at the Campus Tecnológico Local San Carlos, in Santa Clara, Florencia, San Carlos, Alajuela, Costa Rica (CRTM05; X: 444296 Y: 1146016).

### 2.3. Semen Doses, Thawing Procedures, and Study Design

The study used 72 frozen-thawed seminal doses from eight bulls of two breeds (Holstein, n = 4; Jersey, n = 4), using 9 doses from different ejaculates from each animal for analysis. The doses were supplied by three commercial semen importers and two frozen semen suppliers in Costa Rica. During the study period, doses packaged in straws of 0.25 mL volume were used and stored in an MVE^®^ XC 20 Signature (MVE Biological Solutions, Ball Ground, GA, USA) cryogenic tank.

Each dose used was thawed following the procedures described in Valverde et al. [25] in a thermo-regulated laboratory water bath (Digisytem Laboratory Instruments Inc., New Taipei City, Taiwan), using three temperatures (35, 37, and 40 °C) and three thawing times (30, 40 and 45 s). For each thawing temperature, each of the three thawing times was used to examine different combinations of time and temperature for thawing in a 3 × 3 factorial design (nine doses per male and nine time/temperature combinations in which each male was represented once for each combination). After thawing, the contents of each straw were diluted 1:10 (*v*:*v*) with a commercial diluent OptiXcell^®^ (IMV, L’Aigle, France) in Eppendorf^®^ (Sigma- Aldrich, St. Louis, MO, USA, EE. UU.) tubes.

### 2.4. Thermal-Resistance Test (TRT) and Semen Evaluation

All diluted samples were submitted to a TRT consisting of an incubation in an Eppendorf tube at 37 °C for 0.5, 1, and 2 h. After TRT, sperm motility and kinematics variables were analyzed at different post-thawing times. The analysis was conducted using the CASA-mot system (ISAS^®^v1: Integrated Semen Analysis System, Proiser R+D, Paterna, Spain) equipped with a video camera (Proiser 782M, Proiser R+D) that captured images at 50 frames per second (fps) and a final resolution of 768 × 576 pixels. The camera was connected to a UB203 microscope (UOP/Proiser R+D) with a 1× eyepiece and a 10× negative-phase contrast objective (AN 0.25).

The analysis of the samples was carried out according to the procedures used in Víquez et al. [26]. The microscope had an integrated heated stage (37.0 ± 0.5 °C). Subsamples of 3 µL of thawed semen were placed in a Spermtrack^®^ (Proiser R+D, Paterna, Spain) counting chamber that had been pre-warmed to 37 °C. For each of the samples evaluated, two replicates were made. At least seven microscope fields were captured, with a minimum total of 600 sperm cells counted and their motility and kinematics recorded. The values of total motility (TM,%) and progressive motility (PM,%), along with the rectilinear velocity (VSL, μm·s^−1^), curvilinear velocity (VCL, μm·s^−1^), average path velocity (VAP, μm·s^−1^), crossover frequency (beat-cross frequency, BCF, in Hz) and the lateral displacement of the head (ALH, in μm), linearity index (LIN = VSL/VCL·100), straightness index (STR = VSL/VAP·100), and wobble index (WOB = VAP/VCL·100) were used for analyses.

### 2.5. Statistical Analysis

Normality and homoscedasticity were assessed using the Shapiro-Wilk and Levene’s test. Furthermore, the normality of the sperm parameters was evaluated using the normal probabilistic test. Once the hypotheses of normal distribution and homogeneity of variances were tested, a repeated-means ANOVA was performed. Factors within the model, such as breed, temperature, thawing time, and durability over time of thawed semen samples, and the effects of their interactions were evaluated. General linear and mixed models were used to evaluate bovine sperm kinematic variables. Multiple comparison tests were performed with the least squares method, using a Bonferroni correction and a statistical significance level *p* < 0.05. Results are presented as mean ± standard error of the mean. All data were analyzed using the IBM SPSS statistical program, version 23.0, for Windows (SPSS Inc., Chicago, IL, USA).

## 3. Results

The Jersey breed presented higher values of total and progressive motile spermatozoa than the Holstein breed (*p* < 0.05; Table 1). As to spermatozoa movement pattern percentages, the Jersey breed showed a significantly higher rate of fast spermatozoa compared to the Holstein breed (*p* < 0.05). In contrast, for the values (%) of medium and slow spermatozoa, no differences were found between the two breeds.

There were differences between thawing temperatures and sperm motility parameters (*p* < 0.05; Table 2). The highest values for total and progressive motility variables were found when using 37 °C for thawing. Regarding thawing time, a significant effect was found (Table 2) on sperm total motility in relation to the thawing times analyzed.

When assessing motile sperm variables after thawing in the Holstein breed, it was found that both TM and PM decreased (*p* < 0.05) at 2 h (46.12 ± 3.02 and 31.70 ± 2.18%, respectively) when compared to 0.5 h (62.77 ± 2.47 and 43.24 ± 1.49%, respectively). Also, fast spermatozoa and progressive spermatozoa showed a decrease (*p* < 0.05) in the 1 and 2 h periods relative to the 0.5 h period (Table 3).

For the Jersey breed, after thawing, the progressive motile showed significant reductions in the 1 h period in contrast to 0.5h (*p* < 0.05). No differences were found (*p* > 0.05) between times after thawing for total motility and percentages of fast spermatozoa (Table 4).

There was an effect (*p* < 0.05) of breed, thawing temperature, and thawing time on sperm kinematic variables. Interactions were identified between breed and thawing temperature, breed and thawing time, and breed and post-thawing incubation period for all kinematic variables analyzed (*p* < 0.05). When examining the effect of breed, sperm from Jersey bulls showed higher values (*p* < 0.05) in terms of curvilinear velocity (VCL), rectilinear velocity (VSL), average path velocity (VAP), linearity index (LIN), and wobble index (WOB) compared to sperm from Holstein bulls (Table 4). No differences (*p* > 0.05) were observed in the lateral displacement of the head (ALH) between the two breeds. Finally, BCF was higher (*p* < 0.05) in Jersey spermatozoa compared to that in the Holstein breed (Table 5).

After analyzing the overall effect of the thawing temperature of cryopreserved semen on sperm kinematics (Table 6), it was found that the VCL of the doses thawed at 37 °C was higher (*p* < 0.05), followed by those thawed at 35 °C, and finally by straws thawed at 40 °C. When analyzing VSL and VAP, a similar pattern was observed with regard to the temperatures examined. As for LIN, when the thawing temperature was 40 °C, higher values were present compared to temperatures at 35 °C and 37 °C.

When the thawing time of the cryopreserved doses was analyzed, no differences were found (*p* > 0.05) between 30 and 40 s for VCL, but there was a difference (*p* < 0.05) when compared to thawing for 45 s. For the VSL, VAP, LIN, and STR variables, there were no differences between thawing times of 30 s and 45 s. In contrast, differences were found when using 40 s for thawing (Table 7).

The results of sperm kinematics in relation to the post-thawing period revealed significant differences (*p* < 0.05; Table 8). VCL increased as a function of the post-thawing period. VSL and VAP decreased as the post-thawing period was prolonged. These differences are statistically significant and suggest a decrease in semen velocity during the post-thaw incubation period (*p* < 0.05). For LIN, STR, WOB, ALH, and BCF, significant differences in these parameters were observed between post-thawing periods, thus suggesting changes in sperm trajectory over time (*p* < 0.05).

The interaction between the effects of breed, thawing temperature, and thawing time was analyzed. Total motility presented more variation in the Holstein breed than in Jersey breeds, regardless of thawing time and temperature. As the thawing temperature and time increase in both breeds, a decrease in the total motility of sperm (Figure 1).

The interaction between effects of breed, thawing temperature and thawing time for progressive motility also were analyzed. The Jersey breed presented less variation compared to Holstein. The interaction between breed and thawing time showed more variation in the Holstein breed at 40 °C than in the Jersey breed (Figure 2). When analyzing the interaction between temperature and thawing time by breed, the Jersey breed showed higher progressive motility compared to the Holstein breed.

The results of the interaction between the effects of breed, thawing time, and temperature showed high variability. The VCL was higher regardless of the temperature and thawing time in the Jersey breed. In the Holstein breed, there is a downward trend as the temperature and thawing time increase. As to the interaction temperature and thawing time, at 37 °C, there is less difference between the VCL regardless of breed and thawing time (Figure 3).

Analyses of the possible interactions between breed, thawing temperature, thawing time, and post-thawing incubation period were carried out. VCL (Figure 4) showed a lower variation among the velocity values in Jersey bulls for the three temperatures and thawing times analyzed. In the case of the Holstein bulls, a greater variation in VCL values as a function of temperature and time of analysis was observed. When the interaction effect between breed and post-thawing period was analyzed, it was observed that, at 0.5 h after thawing, VCL was lower in relation to longer post-thawing times; however, this phenomenon was inconsistent in the Holstein breed.

The interaction effect between breed and thawing time for VSL showed less variation in the Jersey breed and more variation in Holstein bulls. Moreover, between breed and post-thawing period, there was more variation after 0.5 h for both breeds. However, this variation decreased after 1 h of thawing for both breeds. When analyzing the interaction between breed and thawing temperature, both breeds showed variable values of rectilinear sperm velocity. As to the interaction between breed and post-thawing period, thawed sperm significantly decreased the straight-line velocity after 0.5 h in both breeds, although the effect was more pronounced in the Holstein bulls than in the Jersey ones. In addition, the decrease in the rectilinear velocity of the thawed sperm did not show much variation after 1 h of thawing compared to 2 h post-thawing, an effect that was similar between both breeds (Figure 5).

The interaction between breed and thawing time for VAP showed more variation in the Holstein breed for the three times evaluated. In the case of the Jersey bulls, there was a decrease in this effect. As for the interaction between breed and post-thawing period, it behaved similarly at 0.5 h after thawing in both breeds, with greater values for the Jersey bulls. Furthermore, the interaction between breed and thawing temperature showed higher values of variation at 40 °C and 35 °C in both breeds (Figure 6).

Concerning the interaction between breed and thawing temperature, thawing time, and post-thawing period for LIN, the Jersey breed presented less variation in the three times evaluated, while the Holstein breed did not present the same behavior. With respect to the interaction between the breed and the post-thawing period for both breeds, LIN after thawing at 0.5 h was greater than at the other two post-thawing times. As with the kinematic variables evaluated previously, the interaction between breed and thawing temperature showed more variation in the Holstein breed at 40 °C and 35 °C than in the Jersey breed (Figure 7).

## 4. Discussion

Cryopreserved semen samples from dairy breeds may present variations in sperm motility and kinematics associated with the thawing processes. Previous results obtained by Hirwa et al. [27] in Holstein and Jersey bulls using a visual methodology are lower than those obtained in the present study. However, this allows us to highlight the importance of implementing objective semen analysis methodologies and their relevance to be more precise and not overestimate or underestimate the measurements [28]. The differences observed between breeds could be due to variations among individuals of the same species with respect to post-thaw seminal quality [29] and the freezing and thawing protocols used [30].

It has been shown that variables that describe sperm kinematics vary according to breeds and their production objectives [26]. In Holstein bull, sperm parameters of frozen-thawed semen kinematics have been assessed and found to present variations when compared to other breeds [23]. This suggests a genetic effect on semen kinematics and cell movement patterns in dairy breeds.

Regarding thawing temperature, there is a trend suggesting that the highest values of total and progressive motility are found when 37 °C is used for thawing. This coincides with earlier findings [15] in which a temperature of 37 °C (like body temperature) would provide a favorable environment for more effective metabolic reactivation of frozen sperm to occur. However, it is critical to note that the temperature transition should be gradual to avoid negative effects on sperm cells [24] because thawing temperature influences the sperm kinetic variables of frozen semen. This research supports previous results revealing differences when analyzing different thawing temperatures in frozen straws in Holstein bulls [31]. The higher thawing temperatures could also affect post-thaw sperm motility [30], with an increase in temperature affecting total and progressive motility to a similar extent. Functional damage to the mitochondria can also occur, which negatively impacts ATP synthesis, which, in turn, has a negative effect on motility. This increase in temperature causes the dephosphorylation and activation of GSK3 (glycogen synthase kinase 3), a protein involved in regulating the permeability of the outer mitochondrial membrane, causing a reduction in energy production capacity, so any modification in mitochondrial functioning could manifest negatively on sperm motility [32,33]. In relation to this, our results show an effect of the breed on cellular activity, mainly related to sperm movement after thawing. The Jersey breed suggests a tendency to present greater resistance to cellular damage when compared to Holstein, which is reflected in higher values of total and progressive motility, which became more evident with TRT.

Regarding thawing time, no effect was found on sperm motility in this study, although we observed that when shorter thawing times are used, higher values could be recorded for both total and progressive sperm motility. This is similar to earlier observations [16] showing better sperm motility results at 37 °C for 30 s. Some studies indicate a decrease in the negative effects of recrystallization and hydration on the membrane of the cells; as they are thawed at lower rates, they are in contact for less time with a concentrated solute and the cryoprotectant, so reestablishment of intracellular and extracellular homeostasis will be faster than in the case of slow thawing [17,24]. This agrees with the observation that thawing times affect the kinematics parameters of cryopreserved bovine cells and could also be associated with the temperatures used for thawing, where exposure time at those temperatures could favor heat shock, causing spermatozoon membrane damage [34]. Kinematic variables are important because they describe the swimming patterns of sperm, with those with higher speed and linearity being related to higher fertility rates [35].

The differences in sperm motility between the breeds did not appear when the semen doses were assessed immediately after thawing but rather after a TRT [23]. The results in this study show a significant effect of the thawing period on sperm cell motility and kinematics in both breeds, which suggests that the TRTs can cause observed differences related to the thawing protocol used. This observed pattern coincides with that obtained in a previous work [30], where it was determined that a significant decrease exists in the total and progressive motility of the spermatozoa analyzed after 4 h post-thawing. In addition, there was a decrease in sperm parameters evaluated during a period of 4 h, with thawing temperatures of 36 and 38 °C, respectively, at post-thawing, which was consistent with what was obtained in the present study. In relation to post-thawing sample survival, our results suggest that in some breeds, there could be more predisposition to present more vigorous motility sperm cells in the first hours after semen thawing [30,36]. The results show wide variability in the VCL, VSL, VAP, and linearity according to the TRT and the different thawing protocols; however, at 2 h with the 37 °C for 30 s protocol, these were higher for the majority of variables regardless of the breed. This could be important because it can explain the durability of the semen in the female reproductive tract and its ability to reach the oocyte. The kinematics patterns of Holstein bulls decreased as the post-freezing time increased in a more pronounced way than in Jersey bulls. The evidence found in this work is relevant because it provides more information to validate the possible effects of breed and variations in sperm motility and sperm kinetic parameters, which could be the result of an effect on the individual. Additionally, the interactions of external factors in the post-freezing processes, such as temperature and the time at which the cells are thawed, are conditioning factors of the characteristics that these may present, thus limiting their functionality.

## 5. Conclusions

There was a breed effect on sperm quality at thawing determined by kinematics and cell movement patterns in cryopreserved doses. In addition, the survival effect during post-thaw incubation influences the quality of cryopreserved semen. To ensure sperm viability and functionality, cryopreserved semen doses should be used within 30 min after thawing. Higher values of total and progressive sperm motility were recorded with the thawing protocol of cryopreserved semen doses at a temperature of 37 °C and a thawing time of 30 s. It is important to consider external factors, such as temperature and sperm thawing time, as these influence sperm functionalities. Further research is needed to more accurately understand the behavior of frozen-thawed semen as a predictor of fertility in bovine subspecies.

## Figures and Tables

**Figure 1 animals-14-02768-f001:**
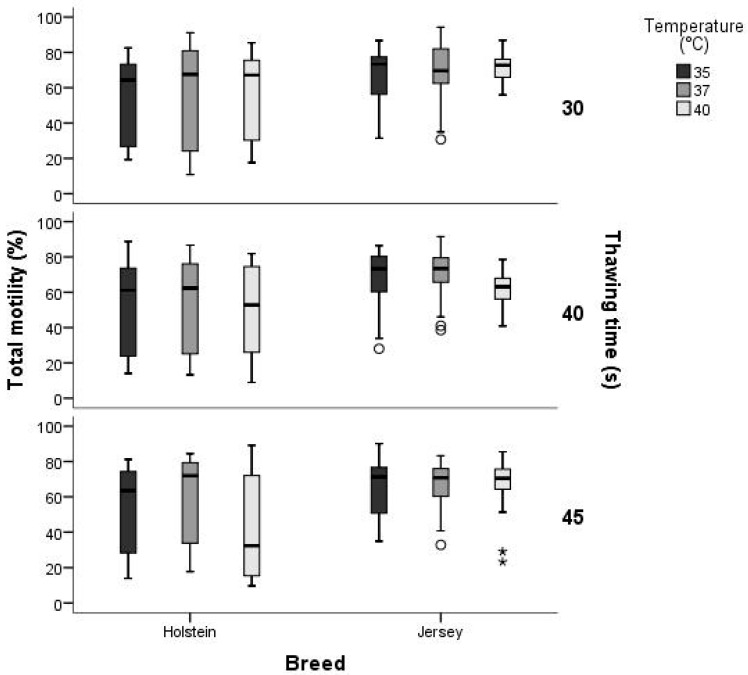
Interaction between breed, thawing temperature, and thawing time for the variable Total motility (TM,%). Area within the boxplot, indicates 50% of the observations between the 25th and 75th percentiles respectively. —: mean value; ┬ ┴: Minimum and maximum values within 3 standard deviation (SD) units.

**Figure 2 animals-14-02768-f002:**
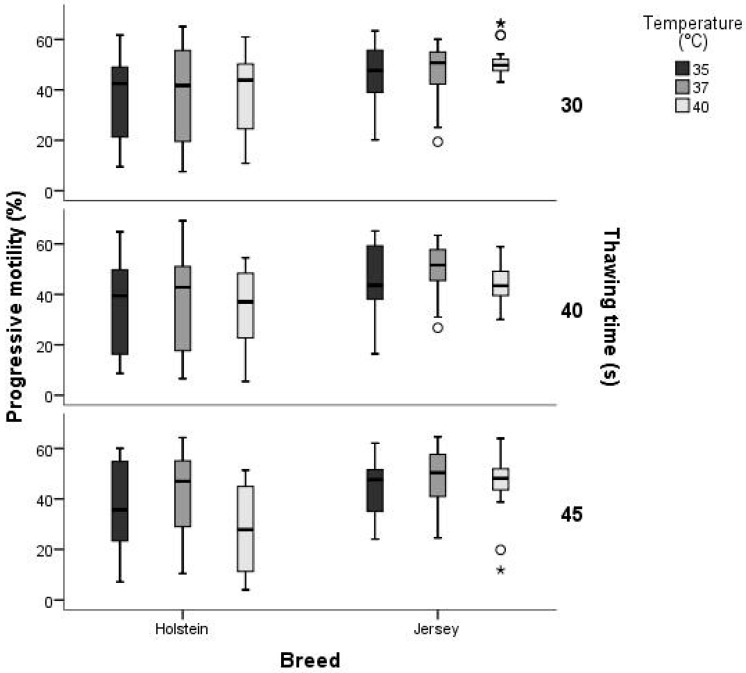
Interaction between breed, thawing temperature, and thawing time for the variable Progressive motility (PM,%). Area within the boxplot, indicates 50% of the observations between the 25th and 75th percentiles respectively. —: mean value; ┬ ┴: Minimum and maximum values within 3 standard deviation units (SD).

**Figure 3 animals-14-02768-f003:**
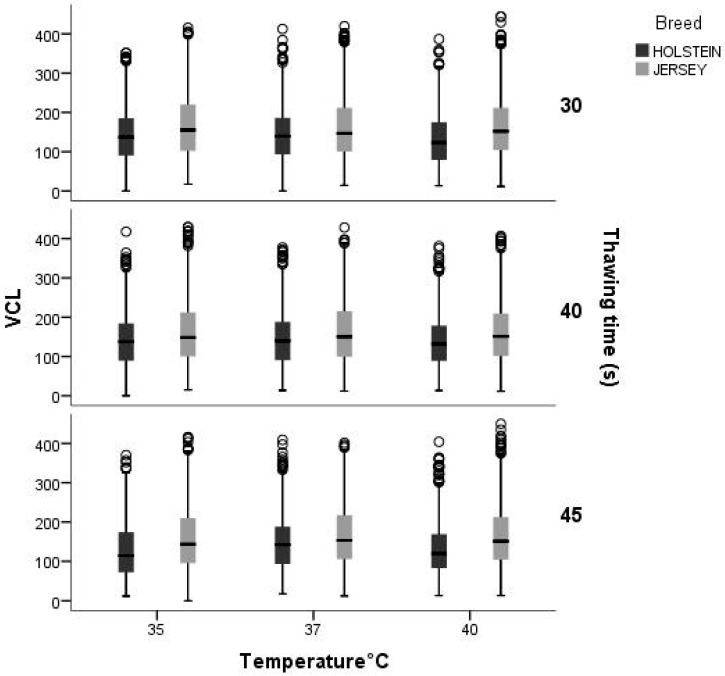
Interaction between breed, thawing temperature, and thawing time for the variable Total motility (VCL, μm·s^−1^). Area within the boxplot indicates 50% of the observations between the 25th and 75th percentiles. —: mean value; ┬ ┴: Minimum and maximum values within 3 standard deviation units (SD).

**Figure 4 animals-14-02768-f004:**
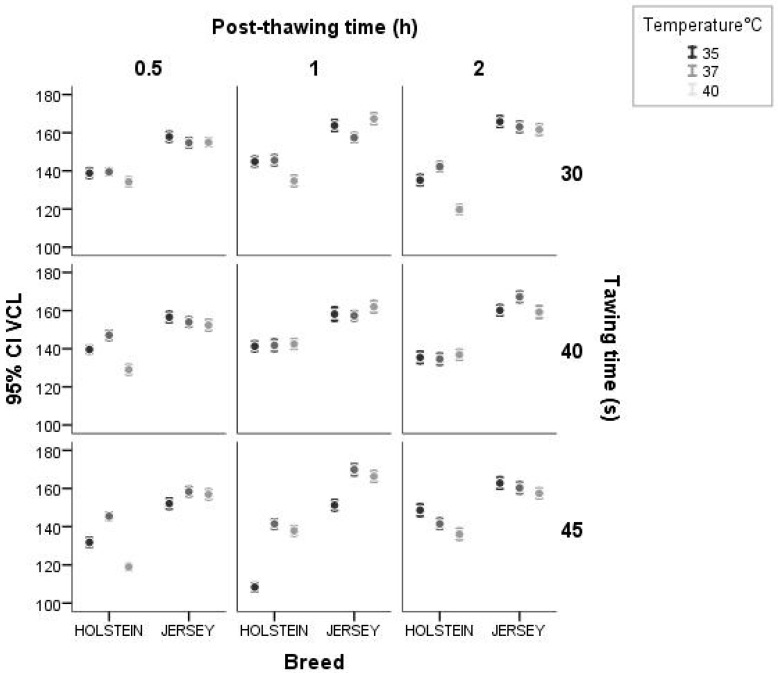
Interaction between breed and thawing temperature, thawing time, post-thawing period for the variable VCL (μm·s^−1^) using a confidence interval of 95% (IC 95%).

**Figure 5 animals-14-02768-f005:**
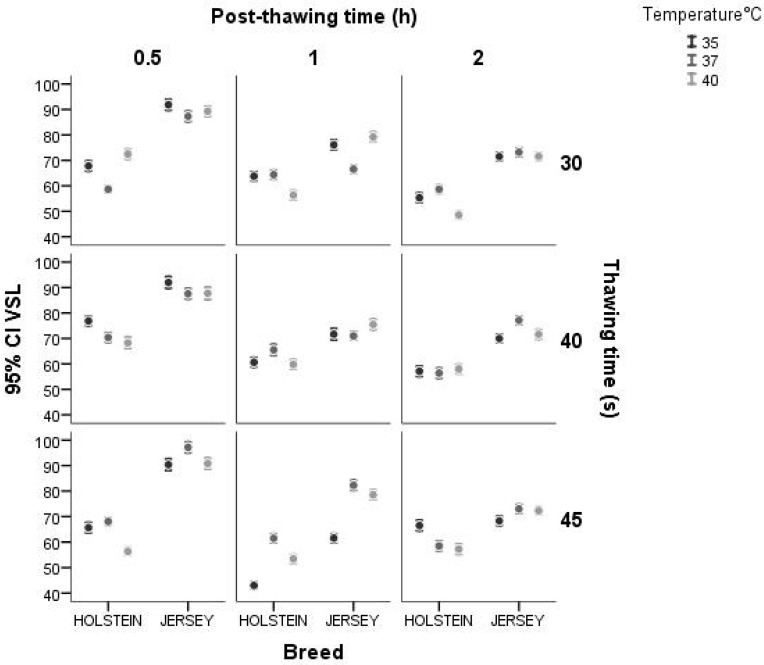
Interaction between breed and thawing temperature, thawing time, and post-thawing period for the variable VSL (μm·s^−1^) using a confidence interval of 95% (IC 95%).

**Figure 6 animals-14-02768-f006:**
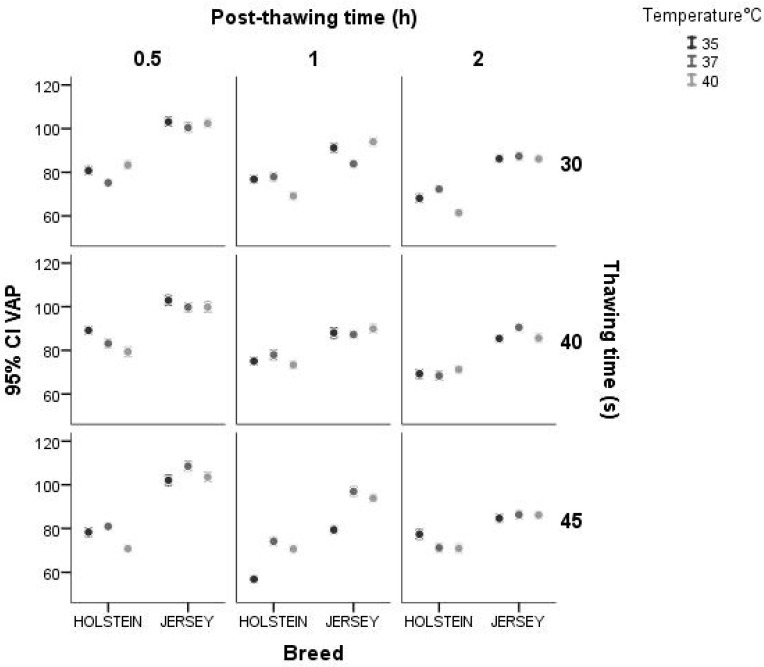
Interaction between breed and thawing temperature, thawing time, post-thawing period for the variable VAP (μm·s^−1^) using a confidence interval of 95% (IC 95).

**Figure 7 animals-14-02768-f007:**
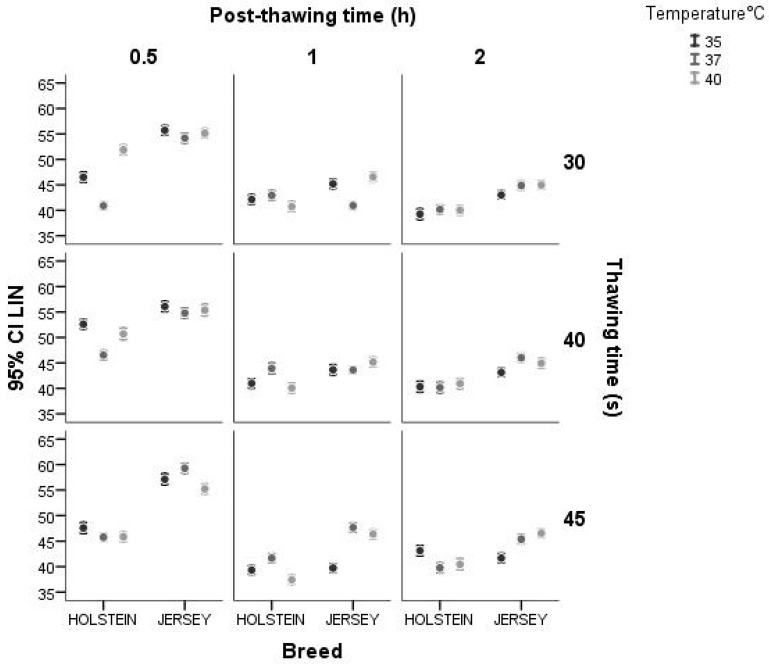
Interaction between breed and thawing temperature, thawing time, and post-thawing period for the variable LIN (%) using a confidence interval of 95% (IC 95).

**Table 1 animals-14-02768-t001:** Estimation of sperm motility (mean ± SEM) in frozen-thawed semen doses from two dairy cattle breeds.

	Breed	
Variable (%)	Jersey	Holstein
TM	64.52 ± 1.45 ^a^	53.10 ± 1.40 ^b^
Fast spermatozoa *	62.38 ± 1.47 ^a^	50.88 ± 1.44 ^b^
PM	45.29 ± 1.00 ^a^	36.30 ± 0.98 ^b^
Medium speed spermatozoa *	1.99 ± 0.12 ^a^	2.21 ± 0.11 ^a^
Slow speed spermatozoa *	0.23 ± 0.03 ^a^	0.20 ± 0.02 ^a^
PM (% from total TM)	70.40 ± 0.72 ^a^	69.68 ± 0.71 ^a^

SEM = standard error of mean. TM = total motility. PM = progressive motility. * Spermatozoa with movement categorized as fast (>45 μm·s^−1^). Different superscripts (^a,b^) indicate differences between breeds. *p* < 0.05.

**Table 2 animals-14-02768-t002:** Evaluation of sperm motility (mean ± SEM) in frozen-thawed semen doses at different thawing temperatures (°C) and times (s) in dairy cattle.

	Variable		
	TM	PM	PM(% from TM)
Temperature (°C)			
35	58.17 ± 1.73 ^a^	39.90 ± 1.25 ^a^	69.82 ± 0.87 ^a^
37	61.32 ± 1.57 ^b^	43.41 ± 1.14 ^b^	70.40 ± 0.87 ^a^
40	58.38 ± 1.61 ^a^	40.66 ± 1.17 ^a^	69.86 ± 0.89 ^a^
Time (s)			
30	61.49 ± 1.61 ^x^	42.63 ± 1.17 ^x^	69.88 ± 0.87 ^x^
40	57.99 ± 1.70 ^y^	40.66 ± 1.22 ^x^	70.14 ± 0.88 ^x^
45	58.40 ± 1.61 ^xy^	40.69 ± 1.17 ^x^	70.09 ± 0.87 ^x^

SEM = standard error of the mean. TM = total motility. PM = progressive motility. ^a,b^ Superscripts with different letters between columns indicate differences between thawing temperatures. *p* < 0.05. ^x,y^ Superscripts with different letters between columns indicate differences between thawing times. *p* < 0.05.

**Table 3 animals-14-02768-t003:** Motile spermatozoa (mean ± SEM) at different times (0.5, 1, and 2 h) after thawing cryopreserved straws from Holstein dairy cattle.

Variables	Post-Thawing Period (h)		
	0.5	1.0	2.0
TM	62.77 ± 2.47 ^a^	50.40 ± 3.23 ^b^	46.12 ± 3.02 ^b^
PM	43.24 ± 1.49 ^a^	33.96 ± 2.10 ^b^	31.70 ± 2.18 ^b^
PM (% from TM)	72.36 ± 1.83 ^a^	69.64 ± 1.24 ^ab^	67.04 ± 1.10 ^b^
Fast spermatozoa *	59.44 ± 2.51 ^a^	48.82 ± 3.26 ^b^	44.39 ± 2.99 ^b^
Medium speed spermatozoa *	2.98 ± 0.18 ^a^	2.20 ± 0.27 ^b^	1.63 ± 0.19 ^b^
Slow speed spermatozoa *	0.35 ± 0.07 ^a^	0.18 ± 0.04 ^b^	0.10 ± 0.02 ^b^

SEM = standard error of the mean. TM = total motility. PM = progressive motility. * Spermatozoa with movement categorized as fast (>45 μm·s^−1^), medium (25 ≤ x < 45 μm·s^−1^), slow (10 < y < 25 μm·s^−1^). ^a,b^ Superscripts with different letters in the same row within Breed indicate differences between times after thawing. *p* < 0.05.

**Table 4 animals-14-02768-t004:** Motile spermatozoa (mean ± SEM) at different times (0.5, 1, and 2 h) after thawing cryopreserved straws from Jersey dairy cattle.

Variables	Post-Thawing Period (h)		
	0.5	1.0	2.0
TM	65.91 ± 2.05 ^a^	65.46 ± 1.31 ^a^	62.29 ± 1.61 ^a^
PM	47.51 ± 1.63 ^a^	43.39 ± 0.86 ^b^	44.76 ± 1.14 ^ab^
PM (% from TM)	71.50 ± 0.72 ^a^	66.94 ± 1.12 ^b^	72.38 ± 0.94 ^a^
Fast spermatozoa *	63.37 ± 2.15 ^a^	63.13 ± 1.62 ^a^	60.71 ± 1.65 ^a^
Medium speed spermatozoa *	2.25 ± 0.14 ^b^	3.46 ± 0.35 ^a^	1.50 ± 0.11 ^c^
Slow speed spermatozoa *	0.29 ± 0.03 ^b^	0.70 ± 0.20 ^a^	0.08 ± 0.02 ^b^

SEM = standard error of the mean. TM = total motility. PM = progressive motility. * Spermatozoa with movement categorized as fast (>45 μm·s^−1^), medium (25 ≤ x < 45 μm·s^−1^), slow (10 < y < 25 μm·s^−1^). ^a,b^ Superscripts with different letters in the same row within Breed indicate differences between times after thawing. *p* < 0.05.

**Table 5 animals-14-02768-t005:** Kinematic variables (mean ± SEM) of frozen-thawed semen doses in dairy cattle.

	Breed	
Variable (%)	Jersey	Holstein
VCL (μm·s^−1^)	156.83 ± 0.27 ^a^	140.07 ± 0.31 ^b^
VSL (μm·s^−1^)	77.37 ± 0.19 ^a^	62.65 ± 0.22 ^b^
VAP (μm·s^−1^)	90.90 ± 0.17 ^a^	76.45 ± 0.19 ^b^
LIN (%)	48.36 ± 0.09 ^a^	42.87 ± 0.10 ^b^
STR (%)	78.58 ± 0.10 ^a^	76.63 ± 0.11 ^b^
WOB (%)	58.66 ± 0.06 ^a^	54.60 ± 0.07 ^b^
ALH (μm)	3.59 ± 0.01 ^a^	3.59 ± 0.01 ^a^
BCF (Hz)	14.38 ± 0.03 ^a^	12.19 ± 0.03 ^b^

SEM = standard error of the mean; VCL = curvilinear velocity; VSL = straight-line velocity; VAP = average path velocity; LIN = linearity of forward progression; STR = straightness index; WOB = wobble index; ALH = lateral displacement of the head; BCF = beat-cross frequency. ^a,b^ Superscripts with different letters in the same row indicate differences between breeds (*p* < 0.05).

**Table 6 animals-14-02768-t006:** Sperm kinematic variables (mean ± SEM) of cryopreserved semen doses in dairy cattle at different thawing temperatures.

Variables	Temperature (°C)		
	35	37	40
VCL (μm·s^−1^)	148.11 ± 0.33 ^b^	151.41 ± 0.32 ^a^	145.84 ± 0.33 ^c^
VSL (μm·s^−1^)	69.80 ± 0.23 ^b^	71.06 ± 0.22 ^a^	69.19 ± 0.23 ^b^
VAP (μm·s^−1^)	83.55 ± 0.21 ^b^	84.73 ± 0.20 ^a^	82.75 ± 0.21 ^c^
LIN (%)	45.36 ± 0.11 ^b^	45.47 ± 0.10 ^b^	46.03 ± 0.11 ^a^
STR (%)	75.96 ± 0.12 ^c^	77.23 ± 0.12 ^a^	76.62 ± 0.12 ^b^
WOB (%)	56.71 ± 0.08 ^b^	56.12 ± 0.07 ^c^	57.07 ± 0.08 ^a^
ALH (μm)	3.61 ± 0.08 ^b^	3.64 ± 0.08 ^a^	3.52 ± 0.08 ^c^
BCF (Hz)	12.85 ± 0.03 ^c^	13.69 ± 0.03 ^a^	13.32 ± 0.03 ^b^

SEM = standard error of the mean; VCL = curvilinear velocity; VSL = straight-line velocity; VAP = average path velocity; LIN = linearity of forward progression; STR = straightness index; WOB = wobble index; ALH = amplitude of lateral head displacement; BCF = beat-cross frequency. ^a–c^ Superscripts with different letters in the same row indicate differences between thawing temperatures (*p* < 0.05).

**Table 7 animals-14-02768-t007:** Sperm kinematic variables (mean ± SEM) evaluated at different thawing times in frozen-thawed semen from dairy cattle.

Variables	Time (s)		
	30	40	45
VCL (μm·s^−1^)	149.07 ± 0.31 ^a^	149.98 ± 0.34 ^a^	147.31 ± 0.32 ^b^
VSL (μm·s^−1^)	69.63 ± 0.22 ^b^	71.12 ± 0.24 ^a^	69.29 ± 0.22 ^b^
VAP (μm·s^−1^)	83.39 ± 0.20 ^b^	84.48 ± 0.21 ^a^	83.16 ± 0.20 ^b^
LIN (%)	45.28 ± 0.10 ^b^	46.04 ± 0.11 ^a^	45.54 ± 0.10 ^b^
STR (%)	76.54 ± 0.12 ^b^	77.10 ± 0.13 ^a^	76.16 ± 0.12 ^b^
WOB (%)	56.28 ± 0.07 ^b^	56.81 ± 0.08 ^a^	56.80 ± 0.07 ^a^
ALH (μm)	3.61 ± 0.01 ^a^	3.58 ± 0.01 ^b^	3.58 ± 0.01 ^b^
BCF (Hz)	13.37 ± 0.03 ^a^	13.41 ± 0.03 ^a^	13.08 ± 0.03 ^b^

SEM = standard error of the mean; VCL = curvilinear velocity; VSL = straight-line velocity; VAP = average path velocity; LIN = linearity of forward progression; STR = straightness; WOB = wobble; ALH = amplitude of lateral head displacement; BCF = beat-cross frequency. ^a,b^ Superscripts with different letters in the same row indicate differences between thawing temperatures (*p* < 0.05).

**Table 8 animals-14-02768-t008:** Sperm kinematic variables (mean ± SEM) in dairy cattle semen analyzed at different post-thawing periods.

Variables	Post-Thawing Period (h)		
	0.5	1	2
VCL (μm·s^−1^)	146.47 ± 0.29 ^b^	149.35 ± 0.33 ^a^	149.53 ± 0.35 ^a^
VSL (μm·s^−1^)	79.17 ± 0.20 ^a^	66.05 ± 0.23 ^b^	64.82 ± 0.25 ^c^
VAP (μm·s^−1^)	91.85 ± 0.18 ^a^	80.77 ± 0.20 ^b^	78.41 ± 0.22 ^c^
LIN (%)	51.70 ± 0.10 ^a^	42.68 ± 0.11 ^b^	42.48 ± 0.12 ^b^
STR (%)	78.23 ± 0.11 ^a^	74.88 ± 0.12 ^c^	76.70 ± 0.13 ^b^
WOB (%)	62.29 ± 0.07 ^a^	54.42 ± 0.08 ^b^	53.19 ± 0.08 ^c^
ALH (μm)	3.38 ± 0.01 ^c^	3.68 ± 0.01 ^b^	3.72 ± 0.01 ^a^
BCF (Hz)	13.20 ± 0.03 ^b^	13.06 ± 0.03 ^c^	13.61 ± 0.04 ^a^

SEM = standard error of the mean; VCL = curvilinear velocity; VSL = straight-line velocity; VAP = average path velocity; LIN = linearity of forward progression; STR = straightness index; WOB = wobble index; ALH = amplitude of lateral head displacement; BCF = beat-cross frequency. ^a–c^ Superscripts with different letters in the same row indicate differences between thawing temperatures (*p* < 0.05).

## Data Availability

The raw data supporting the conclusions of this article will be made available by the authors, without undue reservation.

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
