# Peer review of "Effect of Thawing Procedure and Thermo-Resistance Test on Sperm Motility and Kinematics Patterns in Two Bovine Breeds"

_animals, 2024, doi:10.3390/ani14192768_

Round 1
Reviewer 1 Report
Comments and Suggestions for Authors
Dear authors,
The manuscript is well-written, however, it isn't contribute to new advances in animal reproduction. The results are well-known among academic and comunity target.
Thus, our decision is to reject the paper because it doesn't bring innovation.
Reviewer 2 Report
Comments and Suggestions for Authors
The present study evaluated the effects of thawing time and temperature in combination with a TRT on sperm motility and kinematic variables through a CASA-mot system in Holstein and Jersey bulls. The authors concluded that higher values of total and progressive sperm motility were recorded with the thawing protocol of cryopreserved semen doses at a temperature of 37°C and a thawing time of 30 s. The conceptualization of the study is good, but there are some points that could be improved.
1) Introduction: Clear hypotheses are needed to better justify the research. The authors should emphasize the importance and novel aspects of this work.
2) Material and methods:
a) What was the duration of sperm cryopreservation?
b) Please add reference to the methodologies used.
3) Results:
a) The writing is confusing. It should contain the main relevant findings, and the description of the data and statistical comparisons should be precise.
b) Lines 147-148: “The Jersey breed presented higher values of total and progressively motile spermatozoa than the Holstein breed”. What is the explanation for this?
4) Discussion: emphasize possible practical applications of the results.
5) Conclusions: suggest future studies in the area.
Reviewer 3 Report
Comments and Suggestions for Authors
After carefully reading the manuscript entitled “Effect of thawing procedure and thermo-resistance test on sperm motility and kinematics patterns in two bovine breeds.” I’d like to report my review as follow;
In brief, frozen bull semen from two breeds was thawed with different combinations of temperatures and times. The thawed semen was incubated at 37°C for up to 2 h (thermo-resistance test) and determined for motility and kinematics patterns using a CASA. The results indicated that 37°C for 30 sec was the optimal thawing method.
Strength: the authors utilized thermo-resistance test to determine the best thawing procedure instead of using the results obtained at 0 h post thawed.
Weakness: the best thawing procedure revealed in this manuscript as 37°C for 30 sec is the most popular thawing procedure recommended for frozen bull semen with 0.25 mL straws. Therefore, the results of this manuscript provided no new finding.
Main comments
The experimental design was highly confusing. In the abstract line 33, it was written as the experiments were conducted with a factorial design comprised of three thawing temperatures and three thawing times. As a result, the data were supposed to contain nine treatments (nine thawing procedures). However, in the materials and methods line 110-111, it was stated as “For each thawing temperature, one of three thawing times was used to examine the different combinations of time and temperature for thawing”. Under that circumstance, the thawing procedures were only three procedures. The authors are required to clarify this issue.
With 3x3 factorial designs, the motility results might be presented in the same format as Figure 1 to Figure 4.
If the experiment was carried out as a factorial design, then the variables were 2 breeds x 3 thawing temperatures x 3 thawing times x 3 incubation times = 54 groups as shown in Figure 1 to 4. However, no information of replications was provided.
Specific comments
Simple Summary
Line23-24: motility of Holstein bulls were lower than those of Jersey bulls
Abstract
-Please focus on the effect of thawing procedures or breeds.
-Line 44: please provide more information to support the conclusion that thawing at 37°C for 30 sec was considered to be the optimal procedure.
Introduction
-Please include more references regarding different thawing temperatures and times.
-Please explain more about the advantage of using TRT over 0 h post-thawed to determine the sperm quality.
-Line 81-83: factors affecting CASA results are not relevant to this study.
Materials and Methods
-Please provide information about the provider of the semen.
Results
-To simplify the results, Table 1 could be combined with Table 5. Table 2 could be combined with Table 6 and 7. Table 3 and 4 could be combined with Table 8.
- Why the authors separated results from Holstein and Jersey into Table 3 and 4, respectively but put all results from two breeds for Table 6 to 8.
- Please include statistical determination for interaction, if any, in those Tables.
-Line 150-151: no value of medium and slow in the Table 1.
-Line 152-153: replace “dairy cattle” with “two dairy cattle breeds”
-Table 2: Please rearrange the columns from left to right as TM, PM, and PM (% from TM).
-Table 3: Replace “average speed” with “medium speed”. Please rearrange rows from top to bottom as TM, PM, and PM (% from TM).
-Table 4: same as Table 3.
-Figure 1 to 4: please include the numbers of replication in the figure title.
Discussion
- Please discuss a little more whether thawing at 37°C for 30 sec was the optimal across different incubation times or not.
-Line 302-303: please include some references for “These results”.
-Line 307: the cited reference might not be appropriate because the comparison was made between Holstein and Belgian Blue.
- Line 341-342: How about this study? Any effect of breeds when assessed immediately after thawed.
Conclusion
- Line 363-364: the suggestion to use frozen-thawed bull semen within 30 min might not be relevant since the frozen-thawed semen is intended to be deposited in to cows within 15 min after thawed.
Overall recommendation
Since the manuscript was prepared with an unclear experimental design I’d like to recommend this manuscript as Major Revisions.
Reviewer 4 Report
Comments and Suggestions for Authors
Review of “Effect of thawing procedure and thermos-resistance test on sperm motility and kinematics patterns in two bovine breeds”
The paper presented data on sperm parameters for cryopreserved sperm from Jersey and Holstein bulls at 3 times and temperatures. Additionally, sperm were analyzed up to 2 hours after thawing to determine the impact of being held at 37C. The abstracts were well written; however, there is some English correction needed, especially for the introduction and discussion.
Specific comments below:
Line 40. Spell out what BCF is on first use. The only place in the paper where it was complete spelled out was in Table 5.
Introduction:
Example of english correction needed, Lines 50-52 should read “Reproductive technologies such as artificial insemination (AI) are important practices worldwide for dairy cattle, whose profitability depends directly on their reproductive efficiency.”
Line 57-58. I am not sure what this sentence means. Are sperm resistant to cryopreservation or are they susceptible to damage due to cryopreservation?
Line 72-73 How does this study (citation 14) show critical importance to accurate management?
Line 73-76. This sentence is confusing. Does the rate of cooling affect the rate that it needs to be thawed? That is how it is currently reading. If so, how?
Line 81 “On the other hand” This needs to be changed to a different transition phrase. That is most commonly used to present an opposing thought. Here is it is for two unrelated thoughts.
Line 83 “and others [24]” What are the “others”? Why cite it if it isn’t giving specific information?
Line 85-87. I think this is a massive oversimplification and needs revision. The TRT simulates time at 37 C. It doesn’t simulate conditions of the female reproductive tract, that is way more complex that just a temperature.
Materials and Methods: Specific comments are below, but this needs more details. Were the 9 doses per male from the same or different ejaculates? How many replicates are there for each time/temperature combination? I am assuming that since there were 9 doses per male and 9 combinations of time/temperatures that each male was represented 1 time for each combination. This needs to be explicitly stated.
Line 100 change “done” to “completed”
Lines 108-110. Each dose could not have been thawed using 3 temperatures or times. It should say “…using one of three…..”
Line 114. Samples were diluted in OptiXcell? Was that the media the samples were initially cryopreserved in? Why was this chosen vs a media that does not contain cryoprotectant for dilution?
Statistics- Should a 2 way ANOVA be used to compared between time and temperature?
Results:
This comment carries into the discussion. In your tables 2 and 3, you show significant differences but in the text say there are not differences. I am not sure which is correct but this needs to be consistent. For example. Line 157 says “there were no differences between thawing temperatures on sperm motility parameters” yet in Table 2 there is a difference in PM and TM at 37C compared to the others. Again in line 170, it says there was a difference between time 0.5 and 2 hr but the table indicates there was a difference at 1hr and 0.5 hr as well.
Additionally, in the table, it needs to be stated somewhere that times and/or temperatures are being combined for data analysis (which I am not entirely sure I agree with). I realize in the figures later, there is some documentation for kinetics that shows time and temperature separately but it hard to read and also doesn’t show TM/DM. It would be helpful for the reader to see what the difference is for time 35C-20 sec vs 40C-45 sec. This is all getting lost in this presentation. And it isn’t stated outright that is what is being shown in the table.
Lines 233-242. Why does this section include actual data values found in the Tables when the others do not? They could probably be taken out since there is a table.
In all figures “tawing time (s)” should be “thawing time (s)”. The figure legends are in a grey scale but in the figures it is in color. It is also hard to read. Why not just make it a bar graph with error bars and include statistics?
Discussion:
Overall, this discussion is a little challenging and almost as if the reader is supposed to go grab all the papers cited to get the details. If you are using it to discuss how it relates to your findings, then include the details.
Some things that would be nice to see in the discussion – 1) why do we care about sperm kinetics. What do these measurements tell us or why does it matter that variation in VAP may be higher in Holsteins from a sperm function perspective. 2) Same with TRT. Why does it matter? Previous work by (Vienna et al, 2009) shows that TRT is not associated with fertility outcomes in bulls. Some discussion of why would be relevant.
Line 327. Is GSK3 the protein referred to in line 327? It so, that needs to read “…., a protein involved…..” removing the and. If not, please name the protein.
Line 330. Should this read “ not effect was found”?
Line 335. Should read “……lower rates, they are in contact less time with a…….”
Line 339. This statement is lacking detail clarity. “ This could also be associated with the temperatures used, where exposure…..” Is the temperatures “used” in reference to the paper being cited or this paper. If the paper being cited, include the temperature. In the previous statement, how does thawing time affect the kinematic parameter? Does it make them better or worse?
Lines 342-345. This statement needs rewritten. How can incubating over time cause something to appear at time 0? That is how it is reading currently.
Comments on the Quality of English Language
There are sections of this paper in which the english is very well done. Others in which it needs improved for clarity.
Round 2
Reviewer 4 Report
Comments and Suggestions for Authors
I have rereviewed the paper and my decision would be to accept after minor revision. The authors did a very good job on their revision and addressed all the issues. There is one sentence I would recommend be tweaked. My comments are in quotes below.
“The authors have done a very good job of responding to feedback.
Specific comments:
M&M
Lines 112-114. It would read a little better if changed to “…….using 9 doses from different ejaculates from each animal for analysis.”
Line 122-124. This is much clearer. Great edit!”
Comments on the Quality of English LanguageNot applicable
Author Response
I have revised manuscript based on your comments.
